# OpenReview forum: "The Blind Spot of LLM Security: Time-Sensitive Backdoors Activated by Inherent Features"
_ICLR.cc/2026/Conference — Submitted to ICLR 2026_

### Official Review · Reviewer_nThq · 2025-10-24

**Soundness:** 2
**Presentation:** 3
**Contribution:** 2
**Rating:** 4
**Confidence:** 3

**Summary:**

This paper proposes TempBackdoor, a time-triggered backdoor strategy that uses timestamps in system prompts as a dormant activation signal. The authors build an automated poisoning pipeline (Homo-Poison) and a two-stage training recipe (SFT followed by a focused “n-token” RL) to implant backdoors that only fire when both a future timestamp and a domain condition are present. Experiments on Qwen-2.5 family models report very high attack success, low false positives, and apparent robustness to several existing defenses.

**Strengths:**

- The core idea — using an endogenous system signal (time) as a trigger — is simple but insightful; it exposes a plausible blind spot in certain deployment practices.
- Results are compelling on the controlled Qwen experiments.
- The paper is well written.

**Weaknesses:**

- The attack hinges on the assumption that deployed systems include raw timestamps in model context exactly as trained. Many production stacks sanitize or reconstruct system context server-side (or keep such metadata separate), so the attacker’s assumed access to an unfiltered timestamp field is not convincingly demonstrated. The paper treats “timestamp present” as a binary reality rather than a deployment-dependent variable. This weakens claims about real-world feasibility.

- All evaluations use Qwen check-points and synthetic prompts generated in a tightly controlled pipeline. No experiments on closed APIs, hosted inference stacks, or even a simulation of common sanitization/preprocessing layers are reported. That makes it hard to judge whether TempBackdoor is a lab trick or a practical threat.

- Title and framing promise a broad blind-spot discovery, but the manuscript only operationalizes time. Other supposed “inherent features” (locale, device, region, user-id) are only discussed at a conceptual level. Without experiments showing generalization, the claim that system-level variables broadly form an untested surface is speculative.

- Another limitation is that the paper does not include any comparison with other existing backdoor or trigger designs. Without such context, it’s difficult to gauge how much improvement actually comes from the proposed mechanism rather than from the training pipeline itself.

**Questions:**

- The current Figure 1 is visually useful but the caption and/or markup should explicitly show where the dual triggers are and how they jointly activate the backdoor.

---

> ### Author Response · Authors · 2025-11-21
> **Author Response (part 1 of 2)**
>
> **Dear Reviewer nThq,**
>
> We sincerely appreciate your time and thoughtful feedback, especially the recognition that our core idea is insightful and exposes a plausible blind spot in LLM deployment. We address the concerns as follows.
>
> **W1:**
> >...The paper treats  “timestamp present” as a binary reality rather than a  deployment-dependent variable. This weakens claims about real-world  feasibility.
>
> **A:**
>
> We thank you for this insightful comment. We fully agree that real deployment environments are complex, and that many production systems clean or rewrite the system context or use metadata isolation, rather than passing raw prompts through unchanged.
>
> However, as shown by [1-3] and by our observations of major assistants such as Claude and Grok, keeping some form of temporal information in the context is usually a necessary choice to preserve model utility while enabling basic time-aware reasoning.
>
> Importantly, to test the effectiveness of the attack under non-transparent system prompts, we conduct a timestamp format generalization experiment in Appendix I. The results show that TempBackdoor does not rely on memorizing specific timestamp strings, but instead builds on the model’s abstract semantic understanding of time. Therefore, even if the server rewrites the timestamp format, for example from an ISO representation to a natural language description, the attack remains highly robust as long as the temporal meaning is not completely removed.
>
> **W2:**
> >...That makes it hard to  judge whether TempBackdoor is a lab trick or a practical threat.
>
> **A:**
>
> We understand your concern. When designing our experiments, we explicitly tried to narrow the gap between the lab environment and the real world. We tested whether commercial models in practice include timestamps in their system prompts, and we chose tasks that reflect realistic attack scenarios. In Appendix H, we further conduct a simulation experiment based on real system prompts: we take the released Grok system prompt as a reference, build a system prompt for the Qwen model that is close to a real product setting, and evaluate an already backdoored Qwen2.5-7B-Instruct model on three downstream tasks. The results show that in a setup close to real deployment, our method still achieves high attack success rates and good robustness.
>
> More complex data cleaning procedures in practice are usually applied to user inputs rather than to system prompts. For the aspects mentioned by the reviewer, such as closed APIs and hosted inference stacks, we have added discussion in Section 7 (lines 516-519) to make the paper more complete.
>
> Taken together, our approach has meaningful practical feasibility.
>
> **W3:**
> >Other supposed “inherent features”  (locale, device, region, user-id) are only discussed at a conceptual  level. Without experiments showing generalization, the claim that  system-level variables broadly form an untested surface is speculative.
>
> **A:**
>
> First, in terms of training difficulty, using dynamic timestamps as triggers is clearly more complex than using fixed system variables. Time changes continuously and appears in many different surface forms in the training data. If we still rely on a standard SFT pipeline, it is hard to reliably implant a joint trigger that depends on both time and task domain while preserving the model’s ability to understand time. This is the main reason why we introduce new designs such as Homo-Poison and n-token RL.
>
> Second, to address your concern about whether the attack only works for time and whether it can extend to other system-level features, we also test a setting where a fixed system variable acts as the trigger. Specifically, we use the region field in the system prompt as the trigger condition and, to avoid leaking author information, we set it to Antarctica in the package recommendation task. The results are:
>
> | Metric |  ASR  |  FPR  |  CTA  |
> |:------:|:-----:|:-----:|:-----:|
> |  7B    | 97.1% | 0.9%  | 98.2% |
> |  3B    | 96.9% | 1.1%  | 96.8% |
> | 0.5B   | 93.4% | 3.5%  | 93.2% |
>
> We can see that, compared with using timestamps as triggers, this setting performs better on ASR, FPR, and CTA. This suggests that system-level features such as region can also be used as trigger signals and, under our training framework, can achieve an attack effect comparable to time-based triggers. In the revised version, we have added this experiment and analysis to Appendix K and now state the general conclusion about system-level variables as potential blind spots in a more cautious way.
>
> [1]. Dhingra et al. "Time-Aware Language Models as Temporal Knowledge Bases" TACL 2022.
>
> [2]. Cao and Wang. "Time-aware Prompting for Text Generation" EMNLP 2022.
>
> [3]. Vu et al. "FRESHLLMs: Refreshing Large Language Models with Search Engine Augmentation" ACL 2024.

---

> ### Author Response · Authors · 2025-11-21
> **Author Response (part 2 of 2)**
>
> **W4:**
> >Another limitation is that the paper does not include any  comparison with other existing backdoor or trigger designs. Without such  context, it’s difficult to gauge how much improvement actually comes  from the proposed mechanism rather than from the training pipeline  itself.
>
> **A:**
>
> We thank you for raising this point. It indeed touches on two types of contributions that are easy to mix up: the trigger mechanism itself and the design of the training pipeline. In the current version, we did not separate these two aspects clearly enough.
>
> First, from the design perspective, our goal in proposing a system level backdoor triggered by time is the following. In many real settings, the attacker cannot directly control the user’s input. Traditional backdoors that rely on special trigger words or rare tokens usually require the user to type a specific string, either voluntarily or under strong guidance. This is a strong assumption in real deployments. To relax this assumption, we instead use a built in system variable, the timestamp, as the trigger source. This allows the attack to be reliably activated without any intervention on user inputs, increases its practical relevance, and exposes a security blind spot in current practice where models are often only tested statically at the current time. This part is a conceptual contribution at the level of the trigger mechanism and is largely independent of the concrete training pipeline.
>
> Second, regarding the concern that the performance gains may mainly come from the training pipeline itself, due to the notion of the backdoor trigger is not consistent (it is an abstract logical concept), other backdoor algorithms cannot be compared with ours. Nevertheless, we empirically validate our training scheme: Section 5.3 evaluates the design of the homo-poison strategy, and Section 5.4 compares vanilla SFT and SFT+RL with our proposed SFT + n-token RL pipeline. The results consistently show the effectiveness of our method.
>
> Taken together, these results support the effectiveness of our training pipeline design. In the revised version, we will make it clearer that the training pipeline provides a technical procedure for implanting a high quality backdoor, while the system level trigger mechanism based on timestamps determines whether such a backdoor can open up a new attack surface under realistic evaluation and deployment workflows.
>
> **Q1:**
> >The current Figure 1 is visually useful but the caption and/or  markup should explicitly show where the dual triggers are and how they  jointly activate the backdoor.
>
> **A:**
>
> We thank you for this concrete and helpful suggestion and have revised the manuscript accordingly (see Figure 1, lines 108-122).
>
> Thank you again for your thoughtful feedback. If you have any further questions, we would also welcome the opportunity to engage further.
>
> Looking forward to your response!
>
> Best regards, Authors

---

> ### Author Response · Authors · 2025-11-27
>
> Dear Reviewer nThq,
>
> I hope this message finds you well. With the discussion period nearing its end and approximately seven days remaining, we want to confirm whether our responses have satisfactorily addressed all your concerns. If you have any additional points or feedback you'd like us to consider, please let us know.
>
> Thank you for your time and effort in reviewing our paper.
>
> Best regards, Authors

---

### Official Review · Reviewer_uwv1 · 2025-10-26

**Soundness:** 3
**Presentation:** 3
**Contribution:** 2
**Rating:** 4
**Confidence:** 4

**Summary:**

This paper proposes a poisoning/backdoor attack method based on the system prompt time. By using the time specified in the system prompt as the trigger condition, the method behaves normally for legitimate users before a specific time and produces targeted responses for specific tasks after that time. Experimental results show that the proposed method achieves a high attack success rate across different datasets and models.

**Strengths:**

1. The problem addressed is both novel and important. With the advancement of large language models (LLMs), poisoning attacks that do not require explicit triggers to activate pose new threats to intelligent applications.

2. The evaluation is comprehensive, demonstrating the effectiveness of the proposed method in terms of ASR and robustness.

3. The paper is well organized and easy to follow.

**Weaknesses:**

1. **Lack of theoretical and experimental justification for claimed limitations of existing methods.**
   In the introduction, the paper points out that knowledge-based poisoning attacks lack stealth, but this conclusion is not supported by references, theoretical analysis, or experimental results. In Section 6.1, the paper also fails to evaluate existing knowledge-based poisoning attacks against the adopted defenses. If knowledge-based poisoning attacks are also robust to these backdoor defense strategies, then how can it be proven that they lack stealth?

   Regarding triggerable attacks, the paper claims that attackers cannot control users’ input to activate backdoor attacks. This is reasonable, but I think the proposed method in this paper is more like a knowledge-based poisoning attack under specific conditions, and therefore has different application scenarios compared to triggerable backdoor attacks. It is recommended that the authors focus on knowledge-based poisoning attacks and corresponding defenses in the comparison and evaluation.

2. **The literature review is not comprehensive.**
   As mentioned above, I believe this paper is more aligned with knowledge-based poisoning attacks, yet only one such work (Shu et al., 2023) is discussed without evaluation. It is recommended to introduce and compare the proposed method with more poisoning attacks (e.g., [1–3]).

   - [1] *POISONBENCH: Assessing Language Model Vulnerability to Poisoned Preference Data*
   - [2] *Run-Off Election: Improved Provable Defense against Data Poisoning Attacks*
   - [3] *PoisonedEye: Knowledge Poisoning Attack on Retrieval-Augmented Generation based Large Vision-Language Models*

3. **The proposed attack is easy to defend (according to the stated threat model).**
   The threat model assumes that defenders can detect poisoning attacks without time triggers, which makes it easy to defend against the proposed method—for instance, by simply adding a future timestamp during training.

**Questions:**

Overall, I find this to be an interesting topic with a simple yet effective approach. It is recommended that the authors further clarify the threat model and compare the proposed method with more poisoning attack methods.

---

> ### Author Response · Authors · 2025-11-21
> **Author Response**
>
> **Dear Reviewer uwv1,**
>
> We sincerely thank you for your valuable feedback and for recognizing the novelty of the problem we address as well as the comprehensiveness of our evaluation. Our point-by-point responses to the specific concerns are provided below:
>
> **W1:**
> >Lack of theoretical and experimental justification for claimed limitations of existing methods
>
> **A:**
>
> We apologize for the lack of precision in our original wording. Our intention was not to claim that all knowledge-poisoning attacks are necessarily non-stealthy, but rather to highlight a practical trade-off: when an attacker aims to induce strong and systematic changes in a specific knowledge domain (as in Shu et al., 2023, where the model’s behavior in that domain is heavily altered), this often leads to noticeable shifts or degradation in performance for normal uses in the same domain, making the attack easier for users or evaluators to detect. We agree with the reviewer that our paper should be more careful in positioning itself relative to existing knowledge-poisoning attacks, and we have revised the manuscript accordingly in Section 1 (lines 42-46) and Section 2 (lines 130-134).
>
> **W2:**
> >...I think the proposed method in this paper is more like a  knowledge-based poisoning attack under specific conditions, and  therefore has different application scenarios compared to triggerable  backdoor attacks...
>
> **A:**
>
> In our threat model, we include LLM testers (security auditors) that are common in real deployments. They simulate the questions that users may ask and check whether the answers are correct or in line with  expectations. Knowledge poisoning works in the opposite direction: it  aims to push the model’s behavior on these questions away from the  expected answers, so the conflict between the two is evident.
>
> We agree that the attack proposed in this paper does, in essence, modify knowledge in the target domain, and therefore has many similarities with knowledge-poisoning attacks. What we want to stress is that our poisoning is gated by a system-level time trigger: the activation condition is not decided by an explicit token in the user input, but by the time information in the system prompt. As a result, TempBackdoor is better viewed as a hybrid attack, which
> - in terms of effect, poisons the knowledge in the target domain;
> - in terms of triggering, is activated through the system prompt only when the time t>t0and the domain constraint ω is satisfied.
>
> **W3:**
> >The literature review is not comprehensive.
>
> **A:**
>
> We thank you for pointing out these important works. Our study can be seen as a further extension of prior research on knowledge poisoning, and we are grateful for these contributions. We have expanded the related work in Section 1 (lines 42-49) and Section 2 (lines 130-140) , added the corresponding references, and discussed the three works you mentioned.
>
> **W4:**
> >The proposed attack is easy to defend
>
> **A:**
>
> This is exactly the insight we hope to bring to security teams. With this paper, we aim to refine the current consensus for poisoning attacks. We suggest that testing should go beyond guessing possible user inputs and checking whether the answers match expectations, and should also include testing different system variables to see whether they can cause poisoned behavior in the LLM’s responses.
>
> **Q1:**
> >...further clarify the threat  model and compare the proposed method with more poisoning attack  methods.
>
> **A:**
> We see our main contribution as (to the best of our knowledge) the first to include variables in the system prompt that are not directly controlled by the user into the backdoor threat model, and to extend the traditional threat model on this basis so that it better matches real deployment settings of large language models. Unlike prior work that usually assumes the attacker must control the user’s input to some extent to trigger the backdoor, and even for a very stealthy backdoor it is not easy in practice to make users issue the exact trigger input, we highlight that backdoors can also be activated through system-level context variables, without the user being aware of it.
>
> This reformulated threat model encourages both the LLM security community and companies that operate LLM services to treat backdoor poisoning risks more carefully and seriously. In addition, we use timestamps, a system variable that is widely present in real systems, to design a time-dependent trigger that increases the stealth and impact of the attack vector without changing user inputs, making it possible to bypass current checks that mainly focus on static input content. Finally, we combine several practical techniques for training and fine-tuning LLMs to fully implement this attack vector and evaluate it empirically, showing that this type of threat is practically feasible in real-world environments.
>
> Thank you for the thoughtful feedback. We welcome any further questions or discussion.
>
> Best regards, Authors

---

> ### Comment · Reviewer_uwv1 · 2025-11-25
>
> After reviewing the comments from other reviewers and the authors' responses, I think the authors have largely addressed my concerns. Furthermore, I find it interesting that the authors propose this method not as an attack strategy but as a defensive recommendation. Therefore, I have decided to increase my score.

---

> ### Author Response · Authors · 2025-11-25
>
> Thank you for your positive assessment of our work. If you have any further questions, we welcome you to discuss them with us.

---

### Official Review · Reviewer_rhVw · 2025-10-31

**Soundness:** 4
**Presentation:** 4
**Contribution:** 2
**Rating:** 4
**Confidence:** 4

**Summary:**

The paper introduces a backdoor attack framework against LLMs. The approach is based on training the LLM to be triggered by timestamp features in the system prompt. This allows the attack to be triggered without changing the end-user inputs.

The attack is implemented using an automated data-poisoning method based that is applied during the supervised fine tuning step. The system is trained on poisoned data which is replacing question answer pairs with the same question in a future time and a poisoned version of the same answer.

The system is tested on Qwen 2.5 7B instruct. The system is then activated by the timestamp, which means that it can give different answers to testers then users who access the model later. The backdoor threat would target users who downloaded open source finetuned models or commissioned custom models from third party developers.

**Strengths:**

* Noticing that the timestamp in the system prompt might be a trigger for different responses is an interesting insight. It is unclear whether this insight is novel for this paper. Clearly timestamp based attacks and security techniques have long existed in the literature.
* The authors propose a functional technique to finetune an LLM to provide different answers based on different system prompts.

**Weaknesses:**

* What the approach is creating is basically using any feature in the system prompt to trigger an attack. If the system prompt would have a sentence saying "this is just a test" versus "this is production use", it would be exactly the same thing - and it is also the same thing if they would rely on "Trigger backdoor".
* The fact that the authors managed to train the model to return different results on specific topics based on different system prompts shows a competent training skill, but it is not a major contribution.

* The "performance numbers" of more than 95% are rather meaningless, given that the "attacker" has a complete control over the finetuning.

* What the "Defense" section results really show is that the test-state defence datasets do not test with different timestamps (or at least timestamps that span the timestamp range the authors trained on)

* It appears that it should be very easy to detect the attack by testing with various timestamps or defend it by not using timestamps.

**Questions:**

* Given that it appears to be that this particular attack is easy to defend against, are there other features in the system prompt that might similarly act as a trigger?

**Details Of Ethics Concerns:**

The approach proposes a backdoor attack against LLMs, and thus might provide ideas to an attacker. The fact that the authors expose this risk can provide benefits to the community.

---

> ### Author Response · Authors · 2025-11-21
> **Author Response (part 1 of 2)**
>
> **Dear Reviewer rhVw,**
>
> We sincerely thank you for your time and for recognizing our timestamp-based trigger as an interesting insight and our proposed method as a functional technique. We address the concerns as follows:
>
> **S1, W2:**
> > Novelty and major contributions of this work
>
> **A:**
>
> We view our main contribution as, to the best of our knowledge, the first to include variables in the system prompt that are not directly controlled by the user in the backdoor threat model, and to refine the traditional threat model so that it better matches real deployment settings of large language models. Unlike prior work that usually assumes the attacker must control the user’s input to some extent to trigger the backdoor, we highlight that backdoors can also be activated through system-level context variables, without the user being aware of it.
>
> This reformulated threat model encourages both the LLM security community and companies that operate LLM services to treat backdoor poisoning risks more carefully and seriously. In addition, we use timestamps—a system variable widely present in real systems—to design a time-dependent trigger that increases the stealth and impact of the attack vector without changing user inputs, making it possible to bypass current checks that mainly focus on static input content. Finally, we combine several practical techniques for training and fine-tuning LLMs to fully implement this attack vector and evaluate it empirically, showing that this type of threat is practically feasible in real-world environments.
>
> **W1:**
> > …If the system prompt would have a sentence saying “this is just a test” versus “this is production use,” it would be exactly the same thing….
>
> **A:**
>
> We did not choose the timestamp merely because it is a feature, but because it has three distinctive properties that make it more threatening than an arbitrary string (such as “this is just a test”).
>
> 1. **Functionality.** As you noted, a simple string can be removed by sanitization. In contrast, a timestamp plays an important role in reducing hallucinations and supporting time-sensitive reasoning. Removing it reduces model utility, so “not using timestamps” is not a practical defense. For example, adding the current timestamp to the system prompt can help the model anchor facts in time and avoid outdated or fabricated information. This is a common optimization technique, so using timestamps as triggers is realistic and representative. By contrast, a phrase like “this is production use” typically does not appear in real system prompts (see W5 in the response list).
>
> 2. **Inevitable presence.** Once the system prompt is configured to always include a timestamp, there is no need for the user to provide anything extra.
>
> 3. **Dynamic activation.** Static strings are easy to catch in static code or prompt audits. A future timestamp instead creates a “time bomb”: it allows the backdoor to stay dormant during security review (when the current time is before the trigger time) and then activate automatically after deployment (when time passes the trigger). This dynamic behavior is a core contribution of our work.
>
> **W3:**
> > The “performance numbers” of more than 95% are rather meaningless, given that the “attacker” has complete control over the fine-tuning.
>
> **A:**
>
> Thank you for the comment. We will revise this section to clarify that our comparison with SFT also contrasts two threat vectors: full control of the fine-tuning process vs. poisoning only the training data. This framing makes the experimental comparison more concrete. When the attacker cannot fully control fine-tuning and can only poison data in the training pipeline, the effectiveness is roughly on par with SFT poisoning (see Section 5.4, where plain SFT performs worse). A likely reason is that the trigger is time-based and dynamic, whereas achieving precise knowledge editing via SFT alone is difficult.
>
> Regarding real-world relevance: many developers and companies fine-tune open-source models or commission third parties to deliver fine-tuned models for specific domains [1, 2]. In these settings, a malicious model provider or a third-party fine-tuning vendor can in fact control the entire fine-tuning process. Our attack targets this practical threat. In addition, our threat model is similar to prior “full-control fine-tuning” settings [3, 4], but with a key difference: our trigger relies on system-level variables (e.g., timestamps), so it does not require explicit control of user inputs to activate the backdoor.
>
> [1]. Rahman et al. “HuggingGraph: Understanding the Supply Chain of LLM Ecosystem.” arXiv, 2025.
>
> [2]. Horwitz et al. “Charting and Navigating Hugging Face’s Model Atlas.” arXiv, 2025.
>
> [3]. Huang et al. “Composite Backdoor Attacks Against Large Language Models.” NAACL, 2024.
>
> [4]. Li et al. “Multi-target Backdoor Attacks for Code Pre-trained Models.” ACL, 2023.

---

> ### Author Response · Authors · 2025-11-21
> **Author Response (part 2 of 2)**
>
> **W4:**
> > What the “Defense” section results really show is that the test-state defense datasets do not test with different timestamps (or at least timestamps that span the timestamp range the authors trained on).
>
> **A:**
>
> Thank you for raising this question. We would like to clarify that, in our defense evaluation, the timestamps attached to the test prompts are randomly sampled from the 2026–2027 range, which is the same range used during training, rather than using a single fixed timestamp. We have revised the paper to explain this setting more clearly in Section 6.1.
>
> **W5:**
> > It appears that it should be very easy to detect the attack by testing with various timestamps or defend it by not using timestamps.
>
> **A:**
>
> Your observation is very sharp. This attack can indeed be defended by adding tests that use future timestamps. However, the blind spot that we highlight in this paper is that current standard security evaluations in industry, such as common benchmarks, and typical deployment pipelines almost never include dynamic tests with future timestamps. Our work is to expose this gap and to call on the community to include time based scanning or changes in system prompt variables in standard defense procedures.
>
> At the same time, adding timestamps to the system prompt can effectively reduce hallucinations and improve efficiency at low cost [1-3]. This is why, in our survey, most major platforms already use timestamps. For a deployment, using timestamps in the system prompt is in principle optional, but in practice almost no one refuses such a free gain in performance. Therefore, both in theory and in practice, timestamps are a feasible attack vector.
>
> **Q1:**
> > Given that it appears this particular attack is easy to defend against, are there other features in the system prompt that might similarly act as a trigger?
>
> **A:**
>
> Yes, in Section 6.2 of the previous version of the paper, we already noted that other system prompt features can also pose potential risks and discussed this point to some extent. However, we did not study in depth how practical these other features are as triggers, which weakens the extensibility of our threat model. We now extend our discussion and investigation as follows.
>
> 1. For the open system prompts we examined from Grok and Claude, Claude only includes a timestamp variable. Grok additionally exposes variables such as is_mobile，enable_memory，disable_search，has_memory_management，is_vlm，model_id. Using the elder-plinius/CL4R1T4S repository, we further checked prompts from OpenAI and Gemini and found that they also include information such as region and language.
> 2. From the perspective of evaluators, many of these alternative triggers require stricter conditions. For example, a region based trigger often assumes that the tester and the end user are in different countries, and a language based trigger assumes that they use different languages. These extra assumptions reduce the practical effectiveness of our threat model.
> 3. Using combinations of multiple system features as a joint trigger is also possible. In particular, when the system prompt template and user attributes are both known, such combinations can achieve more precise poisoning while preserving attack effectiveness. This does not change the main conclusion of our work, which is that system prompts can serve as a new threat vector for backdoor attacks.
> 4. To empirically test whether other system-level features can serve as triggers, we use the region field in the system prompt as the trigger condition and conduct additional experiments on the package recommendation task. To avoid leaking author information, we set the region value to Antarctica. The results are:
>
>    | Metric |  ASR  |  FPR  |  CTA  |
>    |:------:|:-----:|:-----:|:-----:|
>    |  7B    | 97.1% | 0.9%  | 98.2% |
>    |  3B    | 96.9% | 1.1%  | 96.8% |
>    | 0.5B   | 93.4% | 3.5%  | 93.2% |
>
>    We observe that, compared with using the timestamp as the trigger, this configuration yields better ASR, FPR, and CTA. This suggests that system-level features (such as region) can also act as trigger signals and, under our training framework, can achieve attack effectiveness comparable to time-based triggers.
> We have incorporated this discussion into Section 6.2 and added the new experiments and analysis to Appendix K.
>
> Thank you again for your thoughtful feedback. If you have any further questions, we would also welcome the opportunity to engage further.
>
> Looking forward to your response!
>
> Best regards, Authors

---

> ### Author Response · Authors · 2025-11-27
>
> Dear Reviewer rhVw,
>
> I hope this message finds you well. With the discussion period nearing its end and approximately seven days remaining, we want to confirm whether our responses have satisfactorily addressed all your concerns. If you have any additional points or feedback you'd like us to consider, please let us know.
>
> Thank you for your time and effort in reviewing our paper.
>
> Best regards, Authors

---

### Official Review · Reviewer_KqSV · 2025-11-10

**Soundness:** 3
**Presentation:** 3
**Contribution:** 2
**Rating:** 6
**Confidence:** 3

**Summary:**

This paper propose a backdoor attack framework for LLMs that exploits timestamp features in system prompts as triggers. The backdoor is activated by a future date then has malicious behavior in specific domains without requiring control over user inputs. It develops an automated pipeline using "Homo-Poison" with a training stragegy combining supervised finetuning and n-token reinforcement learning.

**Strengths:**

S1. The use of system timestamps as backdoor triggers is novel. And the threat model is realisitc that the attackers cannot control user inputs.

S2. Comprehensive experiments evaluation against seven 7 mainstream methods.

S3.Clear paper writting, the paper is will organized and easy to follow.

**Weaknesses:**

W1. The attacker assums that victims will deploy models with timestamp-containing system prompts and this is the basic of successful attacks. The authors should explain why and how the attacker can know this information. Moreover, the attack window limitation between model release and trigger date is mentioned but not thoroughly analyzed.

W2. Although some defense mechanisims (ONION, CUBE) have been evaluated but they were mainly designed for simpler NLP tasks rather than LLMs. The evaluation of defense methods designed for LLM should be evaluated, for example, Random Smoothing.

W3. Lack of model updates or fine-tuning's impact on backdoor persistence.

**Questions:**

Q1. How long backdoors remain effective after the trigger date?

Q2. Have you considered the attack's effectiveness when timestamps are formatted differently across training and deployment?

Others questions please refer to the weakness part.

---

> ### Author Response · Authors · 2025-11-21
> **Author Response (part 1 of 2)**
>
> **Dear Reviewer KqSV,**
>
> We thank you for the careful reading, constructive feedback,  and positive assessment of our novel use of system timestamps as  backdoor triggers, realistic threat model, and comprehensive  experimental evaluation. Below, we address the concerns point by point.
>
> **W1**:
> >The attacker assums that victims will deploy models with  timestamp-containing system prompts and this is the basic of successful  attacks. The authors should explain why and how the attacker can know  this information. Moreover, the attack window limitation between model  release and trigger date is mentioned but not thoroughly analyzed.
>
> **A:**
>
> Thank you for your question about the threat model.
>
> 1. Why and how the attacker knows about the timestamp: Including timestamps in the system prompt has become common practice in industry, because it helps reduce hallucinations and improves the model’s ability to handle time-sensitive queries [1-3]. For model providers, this is a low-cost way to improve performance. Therefore, it is reasonable for the attacker to assume that the victim’s deployment environment will include such a timestamp in the system prompt.
>
> 2. About the attack window: We agree that the attack window is an important practical factor. In third-party agent fine-tuning scenarios, the testing and deployment processes are governed by defined project milestones and quality assurance timelines, which allows the attacker (the fine-tuning entity) to strategically estimate the deployment window. In open-source poisoning scenarios, the attacker can release an updated model version shortly before the trigger date, or use version iteration strategies to shorten the exposure window, thereby reducing the chance of detection before the trigger time is reached. We have added a detailed discussion of this timing strategy in the revised version (Section 7, lines 505-515).
>
> **W2:**
> >...The evaluation of defense methods designed for LLM should be  evaluated, for example, Random Smoothing.
>
> **A:**
>
> Thank you for this helpful suggestion. We would like to clarify one point, which we have already emphasized in the revised manuscript (Section 6.1, lines 455-469). We actually evaluate seven representative defense methods in Section 6.1, not only ONION and CUBE, but also several defenses that are specifically designed for LLMs. In particular, Self-Reasoning uses “defensive examples with reasoning traces” to guide the model to produce normal reasoning and thereby cover and correct backdoored behavior; NAS builds backdoor detection scores from neuron activation statistics to identify and filter suspicious samples; and Shuffle / Shuffle++ apply random shuffling to sentences or words in the reasoning process. In Section 6.1, we now label these methods more clearly as “LLM-oriented defenses” to avoid the misunderstanding that we only consider traditional NLP defenses.
>
> In addition, to further address this concern, we include Random Smoothing in our evaluation. Concretely, we adopt the three perturbation types from SmoothLLM [4] (insert, swap, and patch), apply them to the test set, and fix the overall perturbation rate at 5%, so as to introduce moderate random noise without severely harming task usability. The corresponding attack success rates are:
> | Task Scenario      | Code Generation | Package Recommendation | Food Recommendation |
> |--------------------|:---------------:|:----------------------:|:-------------------:|
> | No Defense         |     96.5%       |         98.6%          |        98.8%        |
> | Random Smoothing   |     88.2%       |         81.4%          |        98.5%        |
>
> As the table shows, under a 5% perturbation rate, Random Smoothing slightly reduces the attack success rate in the code generation and package recommendation scenarios (e.g., from 96.5% to 88.2%), but the success rate remains high overall, and the food recommendation task is almost unaffected (98.8% to 98.5%). This suggests that the perturbations mainly weaken normal semantics, while having limited impact on the trigger pattern tied to the timestamp structure in the system prompt. Therefore, at a 5% perturbation strength that still preserves model usability, Random Smoothing is almost ineffective against TempBackdoor and cannot reliably break the timestamp-based trigger condition.
>
> [1]. Dhingra et al. "Time-Aware Language Models as Temporal Knowledge Bases" TACL 2022.
>
> [2]. Cao and Wang. "Time-aware Prompting for Text Generation" EMNLP 2022.
>
> [3]. Vu et al. "FRESHLLMs: Refreshing Large Language Models with Search Engine Augmentation" ACL 2024.
>
> [4]. Robey et al. “SmoothLLM: Defending Large Language Models Against Jailbreaking Attacks.” TMLR 2025.

---

> ### Author Response · Authors · 2025-11-21
> **Author Response (part 2 of 2)**
>
> **W3:**
> >Lack of model updates or fine-tuning's impact on backdoor persistence
>
> **A:**
>
> We fully agree with your comment and have added corresponding experiments and discussion in the revised version (Appendix J), which show that after incremental fine-tuning on a limited amount of clean data, the TempBackdoor backdoor still maintains a relatively high attack effectiveness. Specifically, starting from a Qwen2.5-7B-Instruct model already backdoored by TempBackdoor, we perform SFT on the package recommendation task using 200, 600, and 1000 clean, task-related samples, respectively. The metrics after fine-tuning are:
> | Setting   |   ASR   |  FPR  |  CTA  |
> |-----------|:-------:|:-----:|:-----:|
> | x1 (200)  |  89.8%  | 1.4%  | 96.9% |
> | x2 (600)  |  76.3%  | 1.3%  | 96.5% |
> | x3 (1000) |  75.6%  | 1.6%  | 96.7% |
>
> We see that as the amount of clean data grows from 200 to 1000 samples, the attack success rate (ASR) does decrease, but it still stays at a relatively high level of about 75%. This shows that, under a typical scale of downstream SFT, the backdoor implanted by TempBack is hard to fully remove. At the same time, FPR remains in a low range around 1%–2%, and CTA stays around 96%, which means the model’s normal-task performance is almost unchanged and the backdoor remains well hidden.
>
> It is important to note that this setting is actually optimistic for the defender: we deliberately use clean fine-tuning data that are from exactly the same distribution as the package recommendation task and contain no poisoned samples. In real code-generation scenarios, defenders usually cannot know in advance which specific domain has been poisoned (for example, a certain type of package recommendation). Their fine-tuning data will be spread across many task types, programming languages, and dependency setups, and its overlap with the backdoor’s target task is often much lower than in our experiment. Therefore, if a victim only performs standard SFT with a limited amount of downstream data on a model that has already been injected with TempBack, the actual reduction in backdoor strength is likely to be even smaller than what we observe here.
>
> In summary, in realistic deployment settings, relying only on incremental fine-tuning with a limited amount of clean data is not sufficient to effectively remove TempBackdoor. The backdoor can still retain a high attack effectiveness and remains robust.
>
> **Q1:**
> >How long backdoors remain effective after the trigger date?
>
> **A:**
>
> In our design, the backdoor is activated whenever the timestamp $t$ satisfies $t > t_0$ and the domain constraint $\omega$ holds. Thus, as long as these two conditions are met, the backdoor remains effective in theory and does not automatically expire on any given date. To study this question, we conduct an “out-of-training-year” extended timestamp test in Appendix E of the originally submitted manuscript: in the package recommendation task, we train the model with timestamps from 2026–2027 and then test it with timestamps from 2028–2039. The results show that the ASR in 2028 is still as high as 96.9%, and 70.3% in 2029. For later years (2030+), the ASR fluctuates and drops, but in some years it can still reach about 15–30%. This indicates that for “near-future” timestamps within several years after the trigger time, the backdoor remains highly stable.
>
> **Q2:**
> >Have you considered the attack's effectiveness when timestamps are formatted differently across training and deployment?
>
> **A:**
>
> Yes, we have already conducted this type of “timestamp format generalization” experiment in Appendix I of the originally submitted manuscript, and we show that TempBackdoor can successfully generalize to timestamp formats that never appear during training. This indicates that the model learns an abstract notion of “time” rather than memorizing a few specific string patterns. During deployment-style testing, we build a separate test set using two unseen timestamp formats. On the package recommendation task, the backdoored 7B model achieves almost the same ASR/FPR on these unseen formats as on the seen ones. This further supports that as long as the string can be interpreted by the model as a valid date/time, the trigger logic can generalize to the new formats.
>
> Thank you again for your thoughtful feedback. If you have any further questions, we would also welcome the opportunity to engage further.
>
> Looking forward to your response!
>
> Best regards, Authors

---

> ### Author Response · Authors · 2025-11-27
>
> Dear Reviewer KqSV,
>
> I hope this message finds you well. With the discussion period nearing its end and approximately seven days remaining, we want to confirm whether our responses have satisfactorily addressed all your concerns. If you have any additional points or feedback you'd like us to consider, please let us know.
>
> Thank you for your time and effort in reviewing our paper.
>
> Best regards, Authors

---

### Author Response · Authors · 2025-11-21
**Summary of Updates**

**Dear All Reviewers,**

We are grateful to all reviewers for their constructive comments. To address them, we have incorporated the following experiments and analyses:

1. **Defense evaluation (Section 6.1):** We added an evaluation of Random Smoothing (Robey et al., 2025), a defense method designed for LLMs.
2. **Persistence evaluation (Section 5.2 / Appendix J):** We study how SFT fine-tuning on different amounts of clean data (200–1000 samples) affects the implanted backdoor.
3. **Generalization evaluation (Section 6.2 / Appendix K):** We added experiments using "Region" (Antarctica) as the trigger. This demonstrates that the attack applies to other system-level variables as well.

In addition, our earlier manuscript already included: a study of the robustness to different timestamp formats (Section 5.2 / Appendix I), an analysis of realistic deployment scenarios based on product-like system prompts (Section 5.2 / Appendix H), and tests with timestamps beyond the training period (Section 5.5 / Appendix E). In this revision, we have updated and streamlined the related descriptions in the main text to present these results more clearly. All modified parts are highlighted in pink-purple for easy reference.

We are truly grateful once again for your feedback.

Thank you, Authors

---

### Meta-Review · Area_Chair_MHPE · 2026-01-10

**Summary:**

This submission received mixed scores (two 4s and two 6s) and was assessed as borderline. We recommend rejection due to remaining concerns from multiple reviewers. The main issues are limited practical realism—stemming from strong assumptions that raw timestamps are preserved in deployed system prompts—and unclear conceptual novelty, as the approach appears closely related to existing knowledge-based or prompt-triggered poisoning methods. As a result, the work is viewed as technically sound but incremental, with split opinions on acceptance. The paper could be strengthened by integrating the rebuttal clarifications into the main text, improving the clarity of the method description, and better contextualizing the experimental evaluation to directly address the reviewers’ concerns.

**Reviewer Concerns:**

The rebuttal successfully addressed several concerns. In particular, Reviewer uwv1 noted that the authors’ clarifications helped better frame the threat model and positioning of the work, and the interpretation of the method as a potential defensive warning rather than purely an attack contributed to a more favorable assessment. Reviewers also agree that, under the stated experimental assumptions, the proposed attack is empirically valid and performs as claimed in controlled settings.

Despite these clarifications, multiple reviewers (KqSV, rhVw, nThq) remain unconvinced about the real-world feasibility of the attack, as it relies on strong assumptions that raw timestamps are preserved in deployed system prompts. Reviewers further argue that the attack appears easy to detect or mitigate, limiting its practical impact, and that the contribution is incremental, being closely related to existing prompt- or knowledge-based poisoning methods. Additional weaknesses include the narrow experimental scope—restricted to limited models and controlled pipelines—and the absence of analysis on backdoor persistence under model updates or further fine-tuning.

**Reviewer Scores:**

Reviewer KqSV: likely unchanged at 6 (marginally above threshold), still viewing the paper as acceptable but weak.

Reviewer rhVw: likely unchanged at 4 (below threshold), maintaining that the contribution is incremental and the attack easy to defend.

Reviewer uwv1: increased score (from 4 → higher, likely 6) after discussion and rebuttal.

Reviewer nThq: linkely unchanged at 4, with persistent concerns about feasibility, scope, and over-claiming.

---

### Decision · Program_Chairs · 2026-01-26

Reject